# Modeling and Analysis of Micro-Grinding Processes with the Use of Grinding Wheels with a Conical and Hyperboloid Active Surface

**DOI:** 10.3390/ma15165751

**Published:** 2022-08-20

**Authors:** Wojciech Kacalak, Filip Szafraniec, Dariusz Lipiński, Kamil Banaszek, Łukasz Rypina

**Affiliations:** 1Faculty of Mechanical Engineering, Koszalin University of Technology, Racławicka 15, 75-620 Koszalin, Poland; 2Doctoral School, Koszalin University of Technology, Racławicka 15, 75-620 Koszalin, Poland

**Keywords:** micro-grinding, modeling, simulation, ceramics

## Abstract

In this article, a method of grinding small ceramic elements using hyperboloid and conical grinding wheels was presented. The method allowed for machining with a lower material removal speed and extending the grinding zone without reducing the efficiency of the process. In order to assess the process output parameters, numerical simulations were carried out for single-pass machining. This strategy allows for automation of the process. Grinding with a low material removal speed is recommended for the machining of small and thin elements, since this can avoid fracturing the elements. The methodology for selecting process parameters as well as the results of the abrasive grains activity analyses were presented. The analyses also concerned the roughness of machined surfaces and the variability of their textures. This grinding method was applied in the production of small ceramic elements that are used in the construction of electronic systems, and in the processing of small piezoceramic parts. This grinding technique could also be used in other grinding processes, where the removal of small machining allowances with high efficiency is required.

## 1. Introduction

The grinding operation is usually the final step in many manufacturing and technological processes. It is used to achieve high dimensional accuracy of parts and high quality of the machined surfaces. The final shape of the processed material is a result of contact between numerous abrasive grains and the workpiece surface; during machining, their exact shape and orientation are random and unknown. During the grinding process, the mechanical and thermal loads as well as chemical reactions affect the tool [1]. These effects lead to wear processes which change the grinding wheel’s active surface properties. On a macro scale, radial changes of the tool can be observed. On a micro scale, grains fracture wear can be distinguished [2]. As a result, an increase in the cutting forces and temperature can be measured, especially in the case of micro wear.

Decreases in the mechanical and thermal loads in the cutting zone can be achieved by making proper choices in machining parameters, or by applying tools with special shapes; these are important considerations, especially for the machining of ceramics. The characteristic mechanical properties of ceramics include low ductility and low fracture toughness. High dimensional accuracy and low surface roughness can be achieved by reductions in the thermal and mechanical loads in the cutting zone [3,4]; fewer fractures will result, especially near the edges of machined parts [5]. Increases in the machining efficiency of ceramic materials are also possible by using a longer cutting trajectory for a single grain. This can be achieved by using a grinding zone of appropriate shape [6], or by using ultrasonic-assisted grinding [6,7].

Grinding with diamond tools is the most often-used method for machining oxide and non-oxide ceramics. Considering the above-mentioned mechanical and thermal loads, machining with conventional abrasive tools would not provide sufficient results for precision grinding. The reason for this is that the depth of cut is greater than the roughness or shape errors of the surfaces before machining.

In the case of finishing machining, especially for large parts with complex geometries, robotic machining centers could be used [8]. For the ceramics grinding process, specialized multi-axis machining centers have been designed with improved dynamic characteristics [9,10]. During the machining of small ceramic elements, the main problem is determining ways to automate the process. An area of base surface that is smaller than 2 cm^2^ can cause difficulties during vacuum fixturing. Small parts with a height of less than 1 mm are also difficult to fasten to the machine table; hence, the parts should be fixed, but that process is not efficient in terms of automation. A solution that has been found is to design work tables that are profiled, and use the grinding force component to keep the parts in position. The highest grinding efficiency that could be achieved using a fully automated process of raw material movement, fixture, removal of the machined parts and single-pass machining strategies. In such a case, a machining strategy that leads to a long contact zone, small infeed and low material removal speed should be applied.

The conducted analyses of the precision machining of ceramics concerned not only the dimensional accuracy of processed parts, their surface flatness and surface integrity [11]; they also included the influence of the machining forces on the mechanical toughness of parts with progressively smaller heights. Important factors in process quality are the geometrical accuracies of the edges and the microstructures of the surfaces. It was found that meeting those requirements is possible through the application of new machining methods and tools. Thus, the results from articles about abrasive tools properties [6,12,13], process kinematics [6,14], the influence of workpiece material properties on the grinding process parameters [15,16,17] and applied coolants [18,19], were taken into account. In preparing the assumptions for the presented grinding strategy, the methodology of process monitoring [20,21] and optimization [22], as well as a detailed analysis of the abrasive tools properties and the topography of the machined surfaces, were included [23,24]. The results of the simulations presented in the study [25,26] were also taken into account. It was found that recently, grinding processes have been applied even to nanoscale machining. The simulations were very helpful for the study of various phenomena related to micro- and nano-grinding [27], since the real-time determination of local values in multiple grinding parameters is difficult.

In this paper, a new method for the face grinding of ceramics, which uses conical and hyperboloid tools, is presented. The axis of the tool was tilted in two perpendicular planes. The α angle corresponded to the perpendicular, and the β angle corresponded to the tangent plane to the feed direction. The β tilt allowed for application of the force in a direction perpendicular to the outer machine ring, which led to the additional keeping effect of parts. The presented method resulted in elongation of the cutting zone. In relation to face grinding with conical or hyperboloid tools without the tilt, the length was multiple times greater. According to peripheral grinding, the resulting contact length was several dozen times longer. As a result, it was possible to ensure low allowance removal speed, high uniformity of the local workpiece load in the grinding zone, the crossing of machining marks and smooth removal of the workpiece from the grinding zone. The advantage of the proposed method is the possibility of achieving efficient material removal process in a single pass.

The method was applied in the production of small ceramic elements used in the construction of electronic systems, in the processing of small piezoceramic parts [6] and in processing valve plates for a water faucet [22]. It could also be used in other grinding processes, where the removal of small machining allowances with high efficiency is required.

## 2. Materials and Methods

### 2.1. Assumptions for Modeling and Machining Simulation

In the presented method for ceramics grinding, the workpiece and tool movements were in the same direction. In the circular motion, the workpiece radius *r_w_* was much greater than the grinding tool radius *r_s_* (*r_w_* >> *r_s_*). Planar face grinding was performed. The active surface of the tool was conical or hyperboloidal. The lowest generatrix of the grinding wheel was at a position parallel to the surface of the workpiece (Figure 1). The axis of the grinding wheel was tilted in two perpendicular planes: at the α angle in the plane perpendicular to the feed direction, and at the β angle in the plane tangent to this direction.

The workpiece trajectory *L*, under the grinding tool, could be described by a curve created on the intersection of cylinder ***w*** with the radius *r_w_*, and the cone or hyperboloid *s* tilted at α and β angles, as shown in Figure 1. The equation for *L* is the following:

L = w(rw) − s(γ) ··· A{w,s},(1)
where γ is the tilt of the tool axis described by α and β angles, *w*(*r_w_*) is the parametric equation for the workpiece trajectory, *s*(γ) represents the parametric equation for the grinding wheel active surface, and $A_{\left\{w,s\right\}}$ is the matrix for transformation grinding tool *x_w_y_w_z_w_* to the grinding wheel *x_s_y_s_z_s_* coordinates.

The grinding wheel active surface with height *h_s_* was described by parametric equations of a cone with base radius *r_s_*, as follows:(2)$$s\left(\gamma \right)=\{\begin{array}{l} x_{s}=\cos\left(t\right)\cdot \frac{v}{tg\left(\gamma \right)}\\ y_{s}=\sin\left(t\right)\cdot \frac{v}{tg\left(\gamma \right)}\\ z_{s}=v \end{array}\;for\;t\;\epsilon \;0÷2\pi ,\;\;v\;\epsilon \;0÷h_{s}$$

And in the case of a hyperboloid, the parametric equations become the following:(3)$$s\left(r_{s}\right)=\{\begin{array}{l} x_{s}=r_{s}\cdot \left(\cos\left(t\right)-v\cdot sin\left(t\right)\right)\\ y_{s}=r_{s}\cdot \left(\sin\left(t\right)+v\cdot cos\left(t\right)\right)\\ z_{s}=v \end{array}\;for\;t\;\epsilon \;0÷2\pi ,\;\;v\;\epsilon \;0÷h_{s}$$

The machined parts were moving along a rotating table with a radius *r_w_*. Their trajectory was described by the parametric equations in the coordinates of the machining table, as follows:(4)$$w\left(r_{w}\right)=\{\begin{array}{l} x_{w}=r_{w}\cdot \cos\left(t\right)\\ y_{s}=r_{w}\cdot \sin\left(t\right)\\ z_{w}=v \end{array}\;for\;t\;\epsilon \;0÷2\pi ,\;\;v\;\epsilon \;0÷h_{s}$$

The transformation matrix between coordinate systems of the grinding machine *x_w_y_w_z_w_* and the grinding tool was the following:(5)$$\left[\begin{array}{c} x_{s}\\ y_{s}\\ z_{s} \end{array}\right]=A_{\left\{w,s\right\}}\cdot \left[\begin{array}{c} x_{w}\\ y_{w}\\ \begin{array}{c} z_{w}\\ 1 \end{array} \end{array}\right]=\left[\begin{array}{ccc} \cos\left(\beta \right) & 0 & \begin{array}{cc} \;\;\;-\sin\left(\beta \right)\;\;\; & 0 \end{array}\\ \sin\left(\alpha \right)\cdot \sin\left(\beta \right) & \cos\left(\alpha \right) & \begin{array}{rr} \sin\left(\alpha \right)\cdot \cos\left(\beta \right) & dy \end{array}\\ \sin\left(\beta \right)\cdot \cos\left(\alpha \right) & -\sin\left(\alpha \right) & \begin{array}{cc} \cos\left(\alpha \right)\cdot \cos\left(\beta \right) & 0 \end{array} \end{array}\right]\cdot \left[\begin{array}{c} x_{w}\\ y_{w}\\ \begin{array}{c} z_{w}\\ 1 \end{array} \end{array}\right]$$
where *dy* is the shift of the origin of the table coordinate system in the 0*y_w_* axis. 

For the above-mentioned transformations, it is possible to acquire a mathematical model for the geometrical parameters of the contact zone between the tool and the workpiece surface (Figure 2).

In order to perform the micro-grinding simulations, the grinding tool active surfaces were modeled according to Equations (2) and (3). The results are presented in Figure 3.

In the simulation process, the parameters of the contact between fragments of the grinding wheel surface and the workpiece cross-sections were calculated. The methodology for their determination is presented in the Figure 3.

### 2.2. Process Simulation Methodology

The mathematical modelling of the face grinding process, especially with conical or hyperboloidal tools, is a complex task due to the relation of contact zone parameters, tilt of the tool’s axes and generatrix location. Another difficulty in designing simulations is the requirement of high computational resources. Relatively to conventional machining, longer contact zones require more calculations. Additionally, the allowance should be removed with low grinding speeds. If the contact length is equal to 70 mm and the depth of cut is 10 µm, the average machining speed is 0.125 µm per 1 mm of workpiece feed. For a specimen moving with a velocity of 2 mm/s and a machining speed equal to 30 m/s (for a tool diameter of *D*_s_ = 250 mm), approximately 10 million grains move over the cutting zone for every millimeter of cutting zone width; in this case, the estimated number of calculations is 10^15^.

The simulation method (Figure 4) allowed for calculations of different kinematic parameters for the process performed with different types of tools. These were achieved by applying procedures for generating contact zones between tool and workpieces on the basis of information about chosen kinematics as well as the type of tool.

## 3. Analysis of the Micro-Grinding Process with the Use of Grinding Wheels with a Hyperboloid Active Surface

### 3.1. Analysis of the Geometric and Kinematic Parameters of the Method

The length of the grinding zone depends on the lowest position of the generatrix of the cone or hyperboloid (Figure 5). The tool tilt angles (α, β) should be selected in relation to the grinding depth and the required length of the contact zone (*L_sn_*). In certain applications of the method [6], when the workpieces were fixed by the grinding forces in a direction perpendicular to the table profile, the length of the grinding zone was slightly smaller than the maximum.

A long grinding zone is necessary to reduce the material removal speed and evenly distribute the load over the workpiece surface throughout the machining process. Additionally, it can be seen that in conventional grinding methods, workpieces are shaped by progressive movements into the machining zone, which limit the possibility of automating such processes.

The presented grinding method ensures good flatness of the machined surface. In the final stage of the process, the material removal speed reduces to 0, indicating that the geometric structure of the machined surface has been improved.

The smallest value of the flatness errors Δ*P* depends on the selected values of the tilt angles α and β, and on the length of the grinding zone *L_sn_*. If the lowest cone or hyperboloid generatrix at the *O_kmax_* point is parallel to the object, the parameter Δ*P* = 0 (Figure 6).

Obtained values of Δ*P* are presented in Figure 6b. These deviations could originate from improper setting of the tilt angles of the grinding wheel axis. Their changes Δα and Δβ from the nominal values could be expressed as a percentage of their respective angle values. An analysis of the presented chart shows that the largest flatness errors (in section A-A, Figure 6a) did not exceed 0.07 μm for the entire width of the object, *b_p_* = 30 mm. The lowest flatness errors (Δ*P*→0) were also observed when the active surface of grinding wheel was set at an improper angle ε*_opt_* (Figure 7).

The methods for selecting the grinding tool tilt angles, presented in Figure 8 and Figure 9, allowed for the longest (most favorable) cutting zone, with a length equal to 98 mm. Hence, the smallest mean value of removed allowance was achieved (*v_h_* = 0.1 µm/mm for *h* = 10 µm).

The proper choice of the axis tilt angles and the position of the lowest generatrix of the grinding wheel active surface (while ensuring the removal of the maximum material thickness *h*) should allow for the lowest average allowance removal speed for which the length of the grinding path *L_sn_* will be the greatest. The value of the *L_max_* parameter depends on the tool diameter *D_s_*, the width of the grinding wheel active surface *b_s_* and the workpiece width *b_p_*. For the parameters *D_s_* = 250 mm, *b_s_* = 40 mm and *b_p_* = 30 mm, the *L_max_* value was 98 mm (Figure 7) That value was almost 2.5 times greater than the grinding wheel active surface width *b_s_*. The ratio will be even greater for the processing of parts that have a smaller width. For example, for an object with a width of *b_p_* = 10 mm, the maximum length of the grinding path *L_max_* will be about 160 mm, which is four times greater than the active surface width of the grinding wheel *b_s_*. This contact length could be even greater for circular grain paths.

Using the chart presented in Figure 8, it was possible to choose the values of process parameters (grinding wheel diameter *D_s_*, width of the active surface of the grinding wheel *b_s_* and parameters of the position and shape of the path of movement of objects) for which the α and β angles allowed for the longest grinding zone *L_sn_*.

Figure 10 shows the maximum values of the allowance height *h* and the corresponding average speed *v_h_* of the material removal as a function of the tilt angle α and the grinding contact length *L_sn_*.

The advantageous feature of the described method is the possibility to obtain very low grinding speeds. For the analyzed data and the average material removal rate, the allowance thickness was *v_h_* = 0.1 μm per one millimeter of workpiece movement. The depth of contact was about 0.032 μm.

### 3.2. Analysis of the Topography Shaping Process for the Machined Surfaces

The presented grinding method, performed by means of conical and hyperboloidal tools, made longer cutting zones possible in relation to conventional machining. Variable directions of traces made by abrasive grains were also observed. The surface textures of machined parts were similar to those where a lapping process is used (crossing grinding marks).

The stages of the machining process performed by means of a hyperboloidal grinding tool are presented in Figure 11. The first stage included initial contacts of abrasive grains and the workpiece surface (third and fourth areas). The second stage was related to engagement of the tool and the workpiece with full contact length (sixth to eighth areas). The third stage concerned removal of the workpiece from the machining zone (10th area), while the machined surface moved underneath the lowest part of the grinding wheel active surface. In that moment, the final roughness was obtained. In the distinguished stages, different texture angles according to the Y axis were observed. The textures crossed at small angles, which was an advantage.

Figure 12 shows the number of active abrasive micro-grains as a function of the grinding length. The activity of their peaks was illustrated for ten different positions of the workpiece in the machining zone during the simulation process. In the first and second micro-grinding stages, the activity of the abrasive grains was low due to the initial entry of the workpiece into contact with the tool, in which the active surface of the grinding wheel was neither flat nor horizontal (Figure 12, zones 1–2). In the following grinding stages, the number of active grains increased to a certain level, along with an increase in the grinding length, and then it stabilized (Figure 12 areas 3–6). In the final stages of the micro-grinding process, a successive decrease in the number of active abrasive grains was observed (Figure 12, zones 7–10). They were able to remove progressively smaller values of the allowance.

### 3.3. Analysis of the Speed of Material Removal and Changes in the Topography of the Machined Surface in the Micro-Grinding Process

In the analyzed micro-grinding method, the material removal speed was variable at a certain moment *t*, and depended on the position of the workpiece in the machining zone. Figure 13 shows the values of the kinematic parameters. Among them, the grinding speed $v_{h}$ could be distinguished. The most favorable values of tilt angles, which achieved the longest cutting path *L_sn_*, were (α < 0, β > 0, ε = ε_p_) for machining performed with a hyperboloidal tool. In that case, surface roughness modelling was carried out. One variant of cutting depth was selected (*h* = 5 µm). After the process, the roughness parameters *Sa* and *St* were calculated. The changes in their values according to the cutting length $L_{sn}$ and the material removal speed $v_{h}$ are presented in Figure 14 and Figure 15. Presented results were approximated by means of a deep neural network. The ratio *St*/*Sa* was also calculated (Figure 16).

The variabilities in selected parameters of the machined surface, as a function of the element’s position in relation to the beginning of the grinding zone (grinding path $L_{sn}$) and the material removal speed $v_{h}$, are presented in Figure 14, Figure 15 and Figure 16.

During the micro-grinding process of the planar surfaces performed with the hyperboloid grinding wheel, three machining stages were distinguished. In the first stage, the grinding wheel removed the largest amount of material (the material removal speed was the highest). After that, only the remaining surface peaks from the previous stage were removed. The value of the *Sa* parameter remained almost unchanged, while the value of the *St* parameter decreased at a constant rate. In the final stage, the ratio of the *St* to *Sa* parameter reached its minimum, and then increased rapidly as the *Sa* parameter decreased.

The increase in the *St*/*Sa* ratio (Figure 16) initiated the second phase in which the value of the *Sa* parameter decreased significantly while the value of the *St* parameter continued to decrease linearly. At this stage of machining, the smoothing of scratches formed during the previous machining began.

The final step was smoothing and creating new surface structure. During this part of the grinding process, the material removal speeds were the lowest. For *h* = 5 µm they decreased from 1.8 µm/s to zero. The material removal speed reached zero when part of the workpiece surface was already below the lowermost generatrix of the tool. In fact, the grinding process was not over. Apart from this position, it was still possible for the higher peaks on the surface to come into contact with the most protruded grains, according to the active surface of the grinding wheel.

The initial material removal speed depends on the selected machining strategy (the position of the lowest forming grinding wheel) as well as the depth of cut and the geometric features of the method (grinding wheel diameter, width of the tool, trajectory of the workpiece path). In the final part of the grinding zone, smooth reduction of the material removal speed to zero provided good conditions for the machining of brittle materials, and resulted in smooth surfaces after single-pass grinding.

## 4. Summary and Conclusions

In this article, the methodology of modelling and simulation of a complex micro-grinding process was presented. The results of the simulations were shown. The parameters that characterized the topography of machined surfaces were indicated. The advantages of using the presented method include the following: a long contact zone, low material removal speeds, small gradients of grain contact depths, high activity of abrasive grains.

On the basis of the conducted simulations, it was shown that the number of active abrasive grains in the proposed grinding method could be as much as 12 times greater compared to circumferential grinding.

Three characteristic grinding zones were defined: (1) the removal of the highest material peaks (the value of the *St* parameter decreased, the value of the *Sa* parameter did not change); (2) smoothing of greater material protrusions of the original workpiece surface (the value of the *Sa* parameter also decreased); (3) forming new surface structures with low levels of roughness. The occurrence of these stages depended on the length of the grinding zone, the grinding depth, the initial surface roughness and the properties of the grinding wheel.

The material removal speed in the proposed micro-grinding method decreased almost linearly for the linear trajectory of the workpieces, and reached 0 at the exit from the machining zone. This ensured an even transition during removal of the workpieces from the grinding zone, as well as reductions in the cross-sections of the cut layers in relation to face grinding (α = 0, β = 0) or grinding with a conical tool (α = 0, β = 0).

The increased length of the grinding zone led to an advantageous surface topography of the machined parts. The main reasons for this were the crossing of the machining traces and a greater number of abrasive grains being involved in this method compared to that in conventional machining.

## Figures and Tables

**Figure 1 materials-15-05751-f001:**
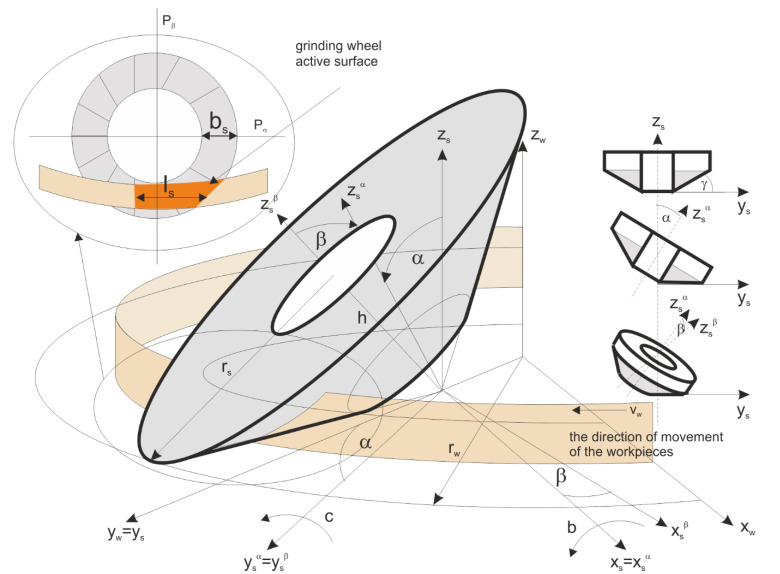
The geometric parameters of the performed grinding method.

**Figure 2 materials-15-05751-f002:**
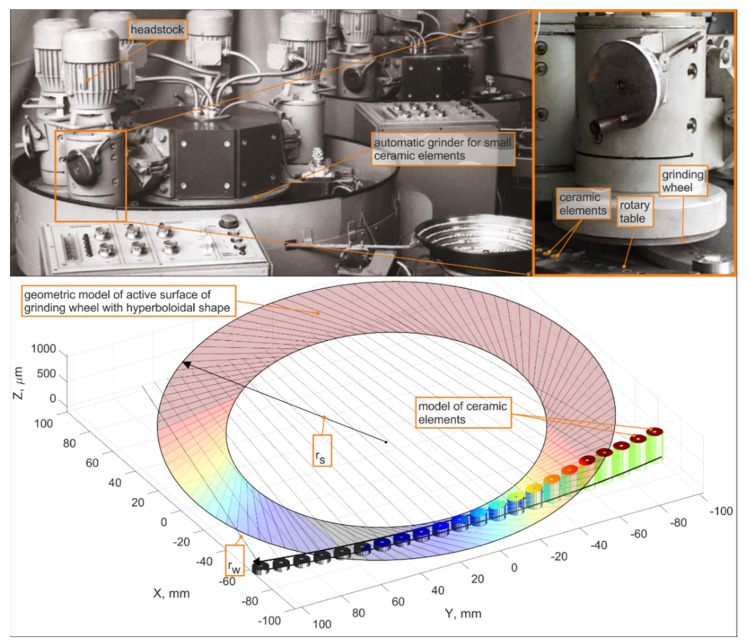
Diagram showing the proposed method of grinding small ceramic elements [6].

**Figure 3 materials-15-05751-f003:**
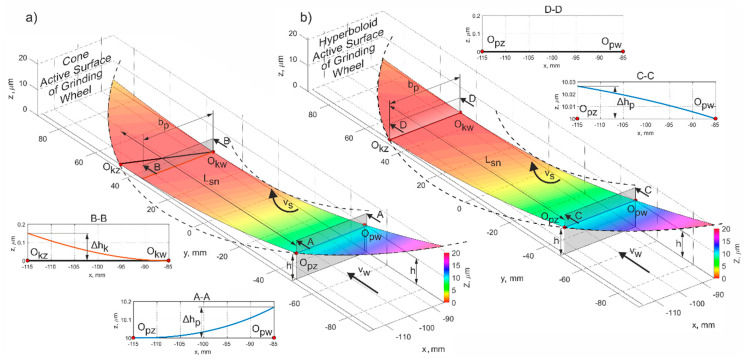
Geometric model of the active surface fragment of the grinding wheel with a conical (**a**) and a hyperboloidal (**b**) shape. *L_sn_*—contact length; *b_p_*—workpiece width; *v_w_*—feed rate; *v_s_*—grinding speed; *h*—depth of cut; Δ*h_p_*—difference of the removed allowance in the initial grinding zone; Δ*_hk_*—difference of the removed allowance in the grinding end zone; *O_pw_* and *O_pz_*—internal and external point of the start of the grinding zone, respectively; *O_kw_* and *O_kz_*—inner and outer point of the grinding zone end, respectively.

**Figure 4 materials-15-05751-f004:**
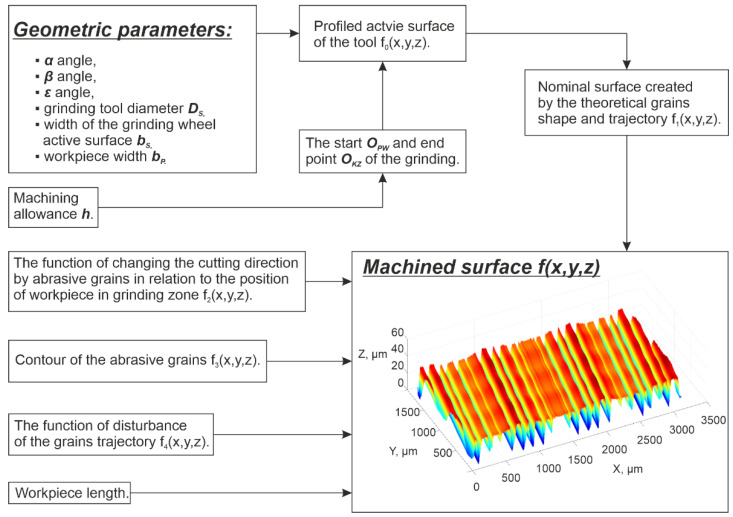
Methodology for generating areas of contact between conical or hyperboloidal tools and the workpiece surface.

**Figure 5 materials-15-05751-f005:**
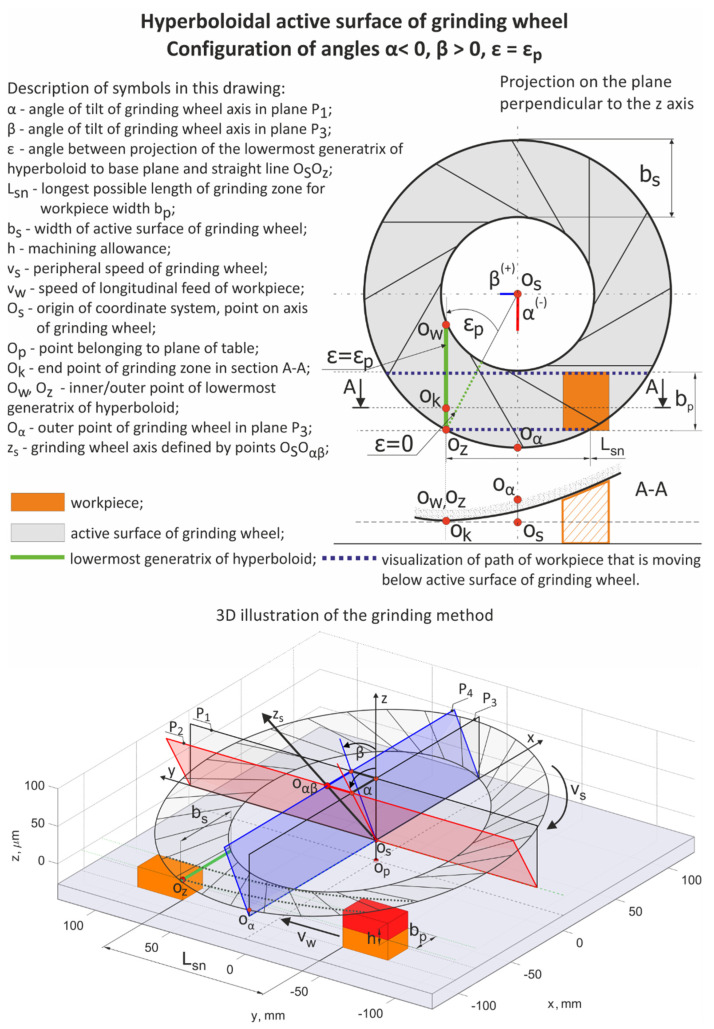
Schematic for the analysis of geometric and kinematic features of the innovative method for micro-grinding surfaces with the face of the grinding wheel.

**Figure 6 materials-15-05751-f006:**
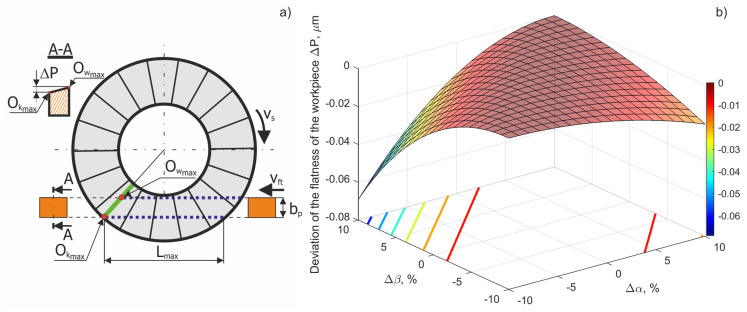
(**a**) Schematic for the surface flatness errors analysis of the workpiece following the micro-grinding process using a hyperboloid active surface. (**b**) Flatness errors of the workpiece Δ*P* as the result of Δα and Δ*β* values.

**Figure 7 materials-15-05751-f007:**
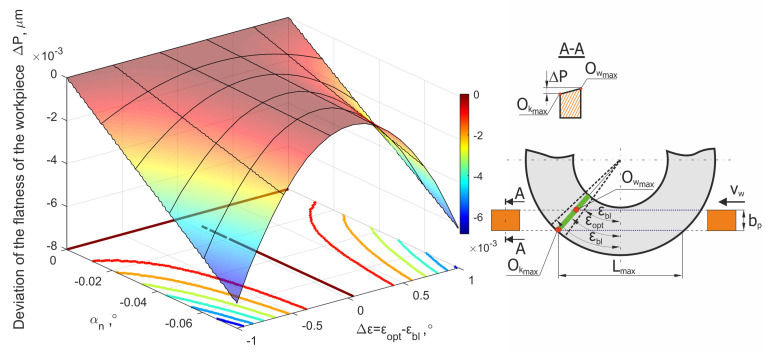
Values of the flatness error Δ*P* as a result of deviations in the tilt angles of the grinding wheel.

**Figure 8 materials-15-05751-f008:**
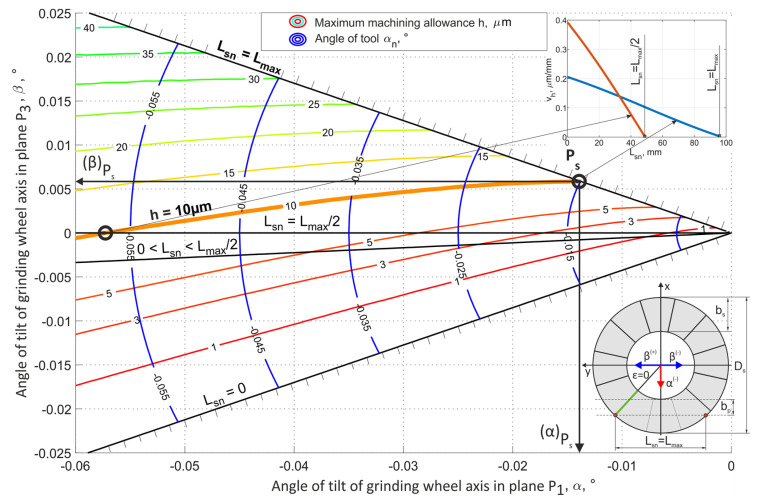
The values of the geometric parameters for the micro-face grinding method using a conical tool; *D_s_* = 250 mm, *b_s_* = 40 mm, *b_p_* = 30 mm.

**Figure 9 materials-15-05751-f009:**
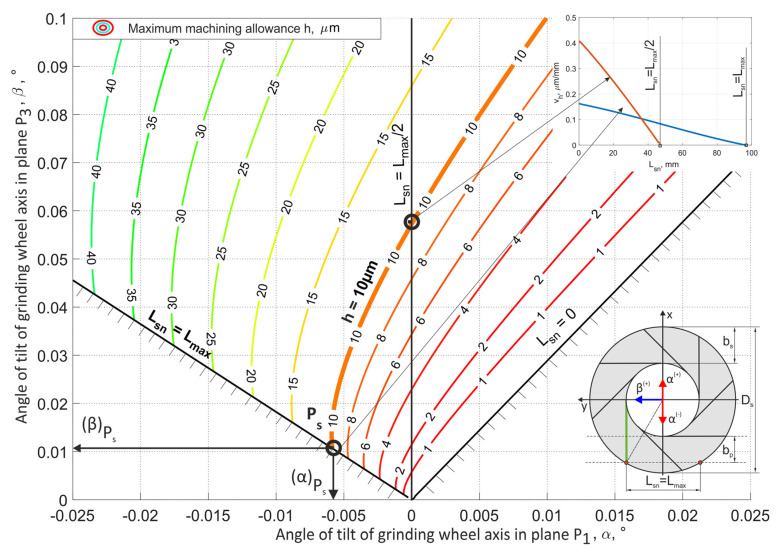
The values of the geometric parameters for the micro-face grinding method using a hyperboloid tool; *D_s_* = 250 mm, *b_s_* = 40 mm, *b_p_* = 30 mm.

**Figure 10 materials-15-05751-f010:**
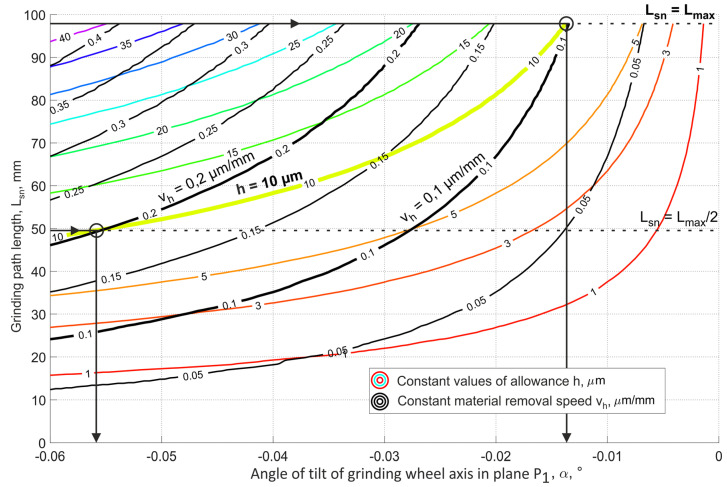
Graph for the analysis of the constant values of the allowance h and the constant material removal speed *v_h_* as a function of the angle of inclination of the axis α and the grinding path length *L_sn_*.

**Figure 11 materials-15-05751-f011:**
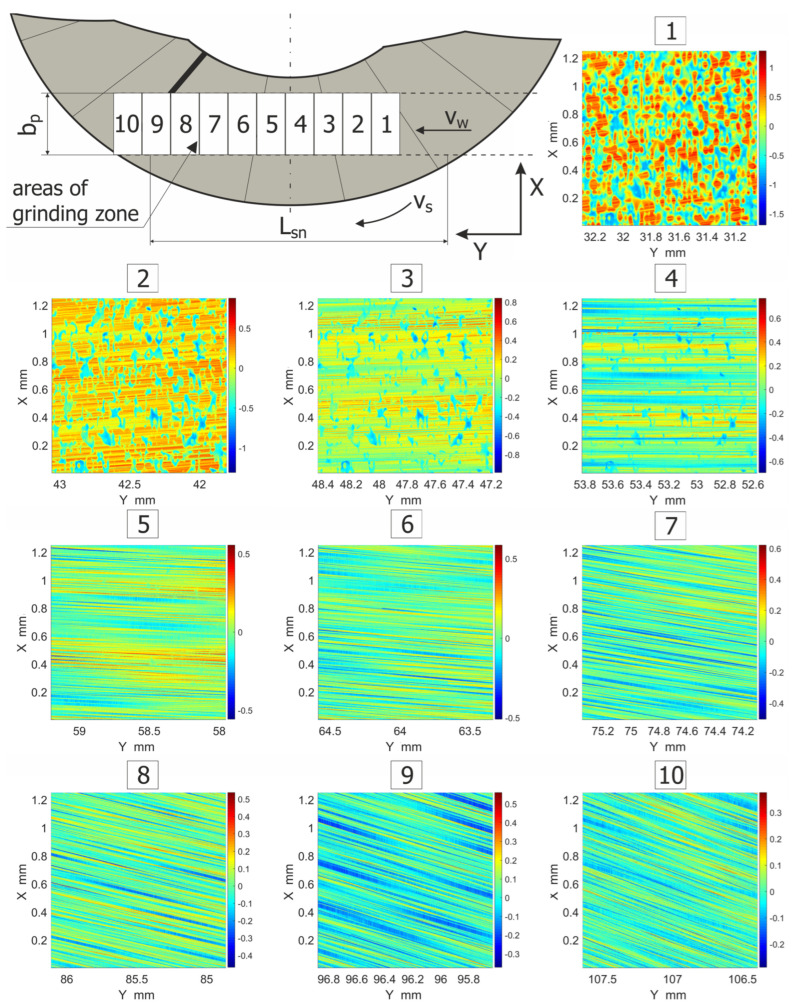
Patterns of the machining traces for selected areas from the ground surfaces.

**Figure 12 materials-15-05751-f012:**
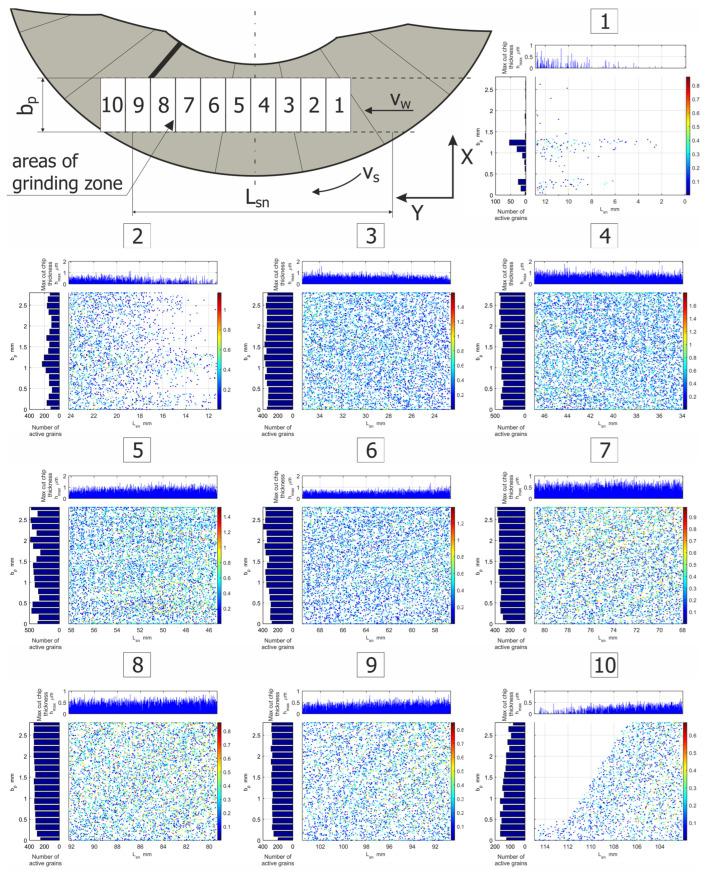
Visualization of the activity of abrasive grains in the micro-grinding process for different positions of the workpiece in the grinding zone.

**Figure 13 materials-15-05751-f013:**
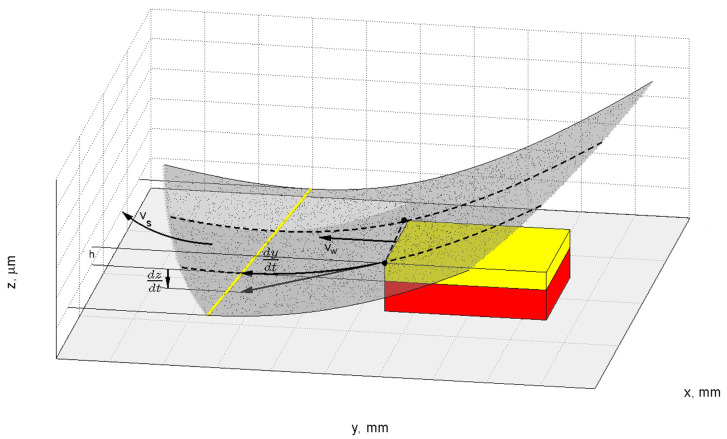
Graphical interpretation of the material removal speed $v_{h}=\frac{dz}{dt}$.

**Figure 14 materials-15-05751-f014:**
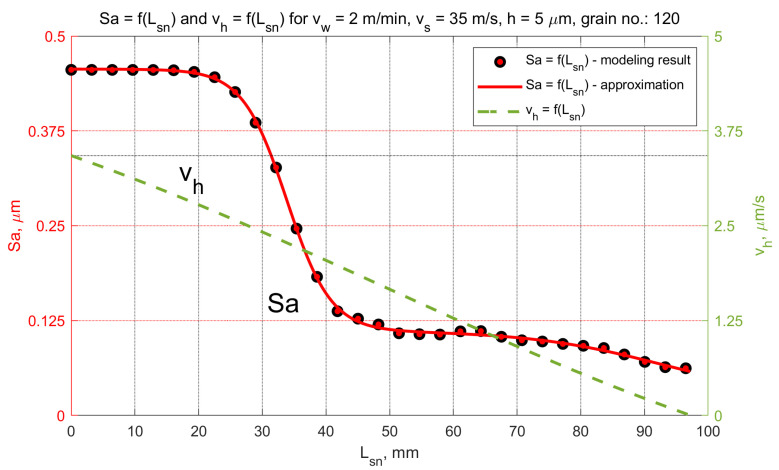
Changes in the *Sa* parameter as a function of the grinding path for a grinding depth of *h* = 5 µm.

**Figure 15 materials-15-05751-f015:**
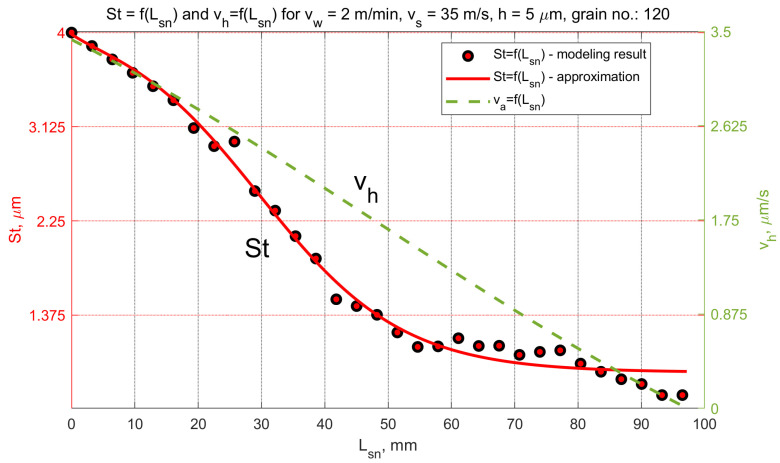
Changes in the *St* parameter as a function of the grinding path for a grinding depth of *h* = 5 µm.

**Figure 16 materials-15-05751-f016:**
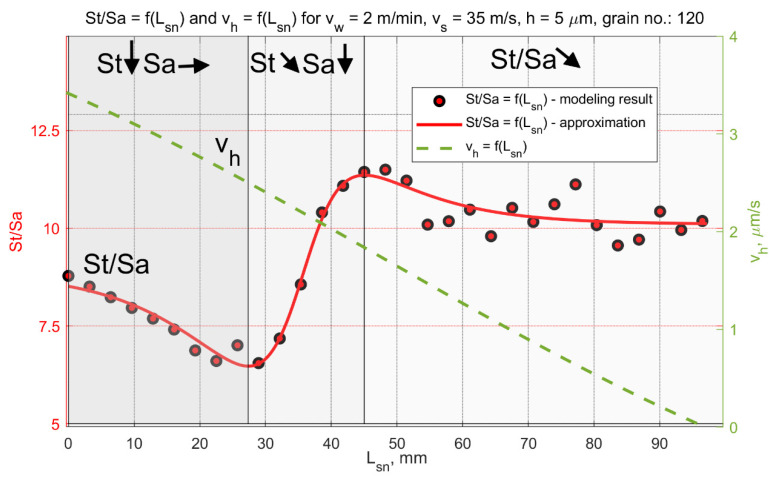
Changes in the ratio *St*/*Sa* as a function of the grinding path for a grinding depth of *h* = 5 µm.

## Data Availability

Data sharing is not applicable to this article.

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
