# Peer review of "Modeling and Analysis of Micro-Grinding Processes with the Use of Grinding Wheels with a Conical and Hyperboloid Active Surface"

_materials, 2022, doi:10.3390/ma15165751_

Round 1

Reviewer 1 Report

This paper presents a method of grinding ceramic parts using hyperboloid and conical grinding wheels. The structure of the paper is clear and reasonable. The research methods and results are valuable. However, there are several questions need be improved and revised. Especially, the English language of this paper needed to be greatly revised. Some specific concerns/comments are listed below:

1. The paper needs proofreading to improve the readability and clarity of the paper, especially the abstract, introduction and conclusion. Please check the reference format carefully.

2. Some of the diagrams need to be supplemented. For example, Figure 2 lacks a description of the X,Y and Z coordinates; It is recommended to add some details in Figures 2, 5, 11, and 12. Pictures need to be sharper and centered; In order to improve the readability of the picture, the annotation of the picture should be clearer and in a reasonable position.

3.Page2, Line 61 and 62, “That could be possible due to the machining strategy allowing......all cutting zone” The expression is not clear and the grammar needs to be fixed. Line 75 and 76, “The conclusion was that in the recent period of application of grinding processes, they were even applied to the nano scale processing.” It is recommended to cite relevant literature.

4. Page 7, Line 177 to 179, “The hyperboloidal shape of the active surface of the grinding wheel starts by creating of the lowest forming of the tool as the horizontal straight line” Need to be revised according to the English expression habits.

5. “3. Summary and conclusions” should be the fourth part of this paper? Please check the titles carefully. It is suggested that the title be modified to make it easier for readers to read.

6.Page14, Line 272 and 273, It is suggested that the ratio of St to Sa should be graphed. What is the significance and value of this ratio?

7. The thesis has done a lot of simulation work, it is suggested to study the effect of hyperboloid and conical grinding wheels in grinding ceramic parts through the combination of simulation and experiment, which improves the value of modeling and analysis results of this paper.

Author Response

Dear Reviewer, thank you for considering our manuscript for publication in Materials. We are grateful for the valuable suggestions provided. In the revised manuscript, all changes are highlighted in red.

Below, we submit responses to the comments to the review.

Point 1: The paper needs proofreading to improve the readability and clarity of the paper, especially the abstract, introduction and conclusion. Please check the reference format carefully.

Response 1: In the manuscript, we have improved the abstract, introduction and summary text. We have adapted the citation format to the requirements of the Materials journal.

Point 2: Some of the diagrams need to be supplemented. For example, Figure 2 lacks a description of the X,Y and Z coordinates; It is recommended to add some details in Figures 2, 5, 11, and 12. Pictures need to be sharper and centered; In order to improve the readability of the picture, the annotation of the picture should be clearer and in a reasonable position.

Response 2: More information has been added to Figure 2. Additionally, the view of the applied grinder for small ceramic elements is presented. The fonts in Figure 5 have been enlarged. The layout of the labels in Figures 11 and 12 has been changed.

Point 3: Page2, Line 61 and 62, “That could be possible due to the machining strategy allowing......all cutting zone”, The expression is not clear and the grammar needs to be fixed. Line 75 and 76, “The conclusion was that in the recent period of application of grinding processes, they were even applied to the nano scale processing.” It is recommended to cite relevant literature.

Response 3: The text on page 2 (Line 61 and 62) has been corrected. The sentence from lines 75 and 76 is a commentary to the quoted references 25 and 26.

Point 4: Page 7, Line 177 to 179, “The hyperboloidal shape of the active surface of the grinding wheel starts by creating of the lowest forming of the tool as the horizontal straight line” Need to be revised according to the English expression habits.

Response 4: That was removed from the manuscript.

Point 5: “3. Summary and conclusions” should be the fourth part of this paper? Please check the titles carefully. It is suggested that the title be modified to make it easier for readers to read.

Response 5: Yes, this should be part 4 of the article. This has been corrected.

Point 6: Page14, Line 272 and 273, It is suggested that the ratio of St to Sa should be graphed. What is the significance and value of this ratio?

Response 6: Figure 16 has been added, showing the "ratio of St to Sa". This dimensionless indicator shows that while the process is in progress, changes to the parameters St and Sa do not occur evenly as the workpiece passes through the processing zone. It indicates three phases of the process: (1) St↓Sa→, (2) St↘Sa↓, (3) St\Sa↘.

Point 7: The thesis has done a lot of simulation work, it is suggested to study the effect of hyperboloid and conical grinding wheels in grinding ceramic parts through the combination of simulation and experiment, which improves the value of modeling and analysis results of this paper.

Response 7: The method was applied for the production of small ceramic elements used in the construction of electronic systems (small capacitors) and the processing of small piezoceramic parts and the valve plates for the water faucet. The results of the experimental research of the proposed method were published in the Materials journal (https://doi.org/10.3390/ma14247904). In the revised manuscript, photographs have been added to Figure 2 with a view of a dedicated grinder for small ceramic elements, together with a view of the grinding wheel and small ceramic elements placed in machining seats on the rotary table.

Moderate English changes required

The text has been revised and the spelling has been corrected.

The authors thank the Reviewer for their time spent analysing the manuscript and valuable comments.

Reviewer 2 Report

1, Please describe the aim of this new method of grinding with hyperboloid and conical grinding wheels, and which kind of ceramic parts need this grinding method.

2.  This paper focus on the simulation of this grinding, so please provide more specific diagram or picture about how the grinding wheel and the workpiece is setup.

Author Response

Dear Reviewer, thank you for considering our manuscript for publication in Materials. We are grateful for the valuable suggestions provided.

The review report stated that our manuscript requires "Extensive editing of English language and style required". We re-edited the text of our manuscript in terms of English language and style, and made changes to the text, especially in the abstract, introduction and summary. The changes made to the revised manuscript are highlighted in red.

Below, we submit responses to the comments to the review.

Point 1: Please describe the aim of this new method of grinding with hyperboloid and conical grinding wheels, and which kind of ceramic parts need this grinding method.

Response 1: “The method was applied for the production of small ceramic elements used in the construction of electronic systems (small capacitors) and the processing of small piezoceramic parts [6] and the valve plates for the water faucet [22]. It could also be used in other grinding processes, where it is required to remove small machining allowances with high efficiency.”
The above paragraph has been moved from Summary to Introduction (Lines 99-102).

Point 2: This paper focus on the simulation of this grinding, so please provide more specific diagram or picture about how the grinding wheel and the workpiece is setup.

Response 2: Pictures showing dedicated grinder for small ceramic elements have been added to Figure 2, along with a view of the grinding wheel and small ceramic elements placed in machining seats on the rotary table.

The authors thank the Reviewer for their time spent analysing the manuscript and valuable suggestions.

Round 2

Reviewer 1 Report

I suggest the paper can be published after some language problems have been revised and improved.